# Constraining models of dominance for nonsynonymous mutations in the human genome

Christopher C. Kyriazis[1]*, Kirk E. Lohmueller[1,2,3]*

**1** Department of Ecology and Evolutionary Biology, University of California, Los Angeles, California, United States of America, **2** Interdepartmental Program in Bioinformatics, University of California, Los Angeles, California, United States of America, **3** Department of Human Genetics, David Geffen School of Medicine, University of California, Los Angeles, California, United States of America

* ckyriazis@g.ucla.edu (CCK); klohmueller@ucla.edu (KEL)

## Abstract

Dominance is a fundamental parameter in genetics, determining the dynamics of natural selection on deleterious and beneficial mutations, the patterns of genetic variation in natural populations, and the severity of inbreeding depression in a population. Despite this importance, dominance parameters remain poorly known, particularly in humans or other non-model organisms. A key reason for this lack of information about dominance is that it is extremely challenging to disentangle the selection coefficient ($s$) of a mutation from its dominance coefficient ($h$). Here, we explore dominance and selection parameters in humans by fitting models to the site frequency spectrum (SFS) for nonsynonymous mutations. When assuming a single dominance coefficient for all nonsynonymous mutations, we find that numerous $h$ values can fit the data, so long as $h$ is greater than ~0.15. Moreover, we also observe that theoretically-predicted models with a negative relationship between $h$ and $s$ can also fit the data well, including models with $h = 0.05$ for strongly deleterious mutations. Finally, we use our estimated dominance and selection parameters to inform simulations revisiting the question of whether the out-of-Africa bottleneck has led to differences in genetic load between African and non-African human populations. These simulations suggest that the relative burden of genetic load in non-African populations depends on the dominance model assumed, with slight increases for more weakly recessive models and slight decreases shown for more strongly recessive models. Moreover, these results also demonstrate that models of partially recessive nonsynonymous mutations can explain the observed severity of inbreeding depression in humans, bridging the gap between molecular population genetics and direct measures of fitness in humans. Our work represents a comprehensive assessment of dominance and deleterious variation in humans, with implications for parameterizing models of deleterious variation in humans and other mammalian species.

**Data Availability Statement:** All scripts are available on GitHub (https://github.com/ckyriazis/dominance).

**Funding:** This work was supported by NIH grant R35GM119856 to KEL from the National Institute

of General Medical Sciences (https://www.nigms.
nih.gov/). CCK and KEL received partial salary
support from R35GM119856. The funders had no
role in study design, data collection and analysis,
decision to publish, or preparation of the
manuscript.

## Author summary

The dominance coefficient ($h$) of a mutation determines its impact on organismal fitness
when heterozygous. For instance, fully recessive mutations ($h = 0$) have no effects on fit-
ness when heterozygous whereas additive mutations ($h = 0.5$) have an effect that is inter-
mediate to the two homozygous genotypes. The extent to which deleterious mutations
may be recessive, additive, or dominant is a key area of study in evolutionary genetics.
However, dominance parameters remain poorly known in humans and most other organ-
isms due to a variety of technical challenges. In this study, we aim to constrain the possible
set of dominance and selection parameters for amino acid changing mutations in humans.
We find that a wide range of models are possible, including models with a theoretically-
predicted relationship between $h$ and $s$. We then use a range of plausible selection and
dominance models to explore how deleterious variation may have been shaped by the
out-of-Africa bottleneck in humans. Our results highlight the subtle influence of domi-
nance on patterns of genetic load in humans and demonstrate that models of partially
recessive mutations at amino-acid-changing sites can explain the observed effects of
inbreeding on mortality in humans.

## Introduction

Dominance is a key concept in genetics, determining the fitness effect of a heterozygous geno-
type compared to that of the two homozygous genotypes. When fully recessive ($h = 0$), a muta-
tion has no impact on fitness in the heterozygous state, and when additive ($h = 0.5$), the fitness
effect of a heterozygote is exactly intermediate to the two homozygous genotypes. The extent
to which mutations have additive or recessive impacts on fitness is critical for many aspects of
evolutionary genetics [1]. For instance, dominance is a central parameter determining the
impact of population size on deleterious variation and inbreeding depression [2–6]. Numerous
studies have demonstrated that the behavior of deleterious mutations in a population greatly
differs when mutations are highly recessive ($h<0.05$) compared to when mutations are par-
tially recessive ($h>0.1$) [2–6]. Thus, estimating dominance parameters is an essential compo-
nent of modelling the effects of demography on recessive deleterious variation.

The importance of dominance in human evolutionary genetics has previously been
highlighted by numerous studies examining the effects of the out-of-Africa bottleneck in
humans on the relative burden of deleterious variation in African and non-African popula-
tions [6–11]. Specifically, these studies aimed to determine whether non-African populations
may have an elevated burden of deleterious variation (also known as "genetic load") due to a
prolonged population bottleneck that occurred when humans migrated from Africa. A general
conclusion that emerged from these studies is that, when deleterious mutations are additive,
recent demography appears to have a slight influence on genetic load, whereas under a reces-
sive model, recent demography may have a more pronounced effect on genetic load [6–10].
These findings have important implications for our understanding of how human evolution-
ary history has impacted the efficacy of natural selection across human populations.

Despite this significance of dominance in evolutionary genetics, dominance parameters
remain very poorly quantified in humans and other vertebrates. In humans, almost no esti-
mates of dominance parameters exist, and those that are available are for small subsets of the
genome. For instance, Balick et al. [6] devised an elegant approach of contrasting patterns of
genetic variation in bottlenecked vs. non-bottlenecked populations, based on the knowledge
that bottlenecks impact genetic variation in differing ways when mutations are additive or

recessive [5]. Their analysis estimated that genes associated with known autosomal recessive diseases in humans have an average dominance coefficient of $h = 0.2$. However, the degree to which this estimate applies to a broader set of deleterious mutations remains unclear. In a follow-up study, these authors also attempted to identify genes under recessive selection in the human genome from allele frequency data [12]. Their analysis demonstrated that, although there is little power to identify individual genes under recessive selection, such genes can be detected in aggregate. Thus, although some recent progress has been made in furthering our understanding of dominance in humans, numerous fundamental aspects remain poorly understood.

In a laboratory setting, several experimental studies have been conducted aiming to quantify dominance using model organisms such as *Drosophila melanogaster* and *Saccharomyces cerevisiae*. These studies have generally found support for deleterious mutations being partially recessive, with mean $h$ estimates ranging from ~0.1–0.4 [13–18]. Moreover, some studies have also found support for a relationship between $h$ and $s$ (hereafter, *h-s* relationship), where more deleterious mutations tend to be more recessive [14,17,18]. This negative relationship between $h$ and $s$ was predicted by theoretical models of dominance proposed by Wright and Haldane [19,20]. However, there are many caveats associated with these experimental studies. First, experimental manipulations are time-consuming and only a handful of such experiments have been conducted. Consequently, several studies have reanalyzed existing experimental data and found that estimates of $h$ may depend greatly on the analytical approach and that such estimates are typically associated with a fair amount of uncertainty [14,15]. Moreover, given that these experiments have been conducted on model organisms in controlled laboratory settings, it remains unclear whether these results are relevant for natural populations in vertebrate taxa such as humans.

Another common approach for estimating selection and dominance parameters relies on using evolutionary models to infer parameters from patterns of genetic variation, as summarized by the site frequency spectrum (SFS; reviewed in [21]). Based on the Poisson random field model [22], Williamson et al. [23] developed an approach to infer dominance using diffusion theory to model the change in allele frequency over time due to genetic drift and selection. Parameters, including $h$, are estimated by finding the values that yield a SFS that is close to that from the empirical data. While this approach is theoretically elegant, it has not been applied to many species. One reason for this is that, when the distribution of fitness effects for new mutations (DFE) is unknown, it is hard to distinguish between different combinations of $s$ and $h$ [24,25]. Intuitively, most deleterious mutations in natural populations are segregating in the heterozygous state. This provides information about $h^*s$, rather than $s$ and $h$ separately [24,25]. Given these challenges, studies that have attempted to estimate the DFE typically ignore dominance entirely by assuming that all mutations are additive (e.g., [26–29]). However, in humans, one previous study attempted to fit non-additive models to the SFS, assuming the same value of $h$ for all mutations [30]. This study found that a range of $h$ values fit the data reasonably well, so long as $h > 0.3$ [30]. However, this study did not attempt to fit an *h-s* relationship, given the challenges associated with identifiability of $h$ and $s$ parameters.

Previously, Huber et al. [25] circumvented this identifiability issue by leveraging selfing and outcrossing *Arabidopsis* populations for estimating dominance parameters. Genetic variation data from the outcrossing population provided information about $h^*s$ while the selfing population provided information about $s$, because all mutations were found in the homozygous state. Using this framework, Huber et al. [25] found statistical support for an *h-s* relationship for amino acid changing mutations in *Arabidopsis*, with more deleterious mutations being more recessive. Specifically, they found that even moderately deleterious mutations ($0.001 < |s| \leq 0.01$) were highly recessive ($h < 0.05$), whereas more neutral mutations may be dominant

($h$ = 1) or additive ($h$ = 0.5) [25]. These results contrast with previous experimental work, where only very strongly deleterious mutations ($|s|$>0.1) are typically found to be highly recessive ($h$<0.05) [13–18]. Moreover, these results also contrast with previous SFS-based studies in humans, where highly recessive models were shown to have a poor fit to the data [30]. The extent to which these contrasting results may be due to methodological reasons or true differences in dominance parameters across species remains unclear.

In the relative absence of information about dominance in humans or vertebrates, many studies instead opt to use *ad hoc* dominance parameters for modelling deleterious variation (e.g., [3,7]), or explore only the extreme cases of additive and fully recessive mutations (e.g., [10,31]). For instance, Henn et al. [7] formulated an *h-s* relationship for modelling genetic load in humans that was highly recessive, where *h* declines below 0.05 when $|s|$ increases beyond ~0.001. This *h-s* relationship parameterization has since been employed by a number of other studies [3,32–34], though a recent analysis showed that such a highly recessive model is not consistent with measures of inbreeding depression in humans [35]. By contrast, other recent studies have employed much less recessive dominance parameters for modelling genetic load in wild mammals [36,37], where *h* declines below 0.05 only as $|s|$ increases beyond ~0.2. However, these parameters have also shown to be inconsistent with available evidence in humans and other mammals [35]. Thus, a very basic practical need for parameterizing evolutionary simulations is a set of realistic selection and dominance parameters that are directly inferred from genetic variation datasets.

Here, we explore the fit of selection and dominance models to patterns of genomic variation in humans. Given that we cannot reliably separate *s* and *h* based on genetic variation data, we instead constrain the range of dominance models that are consistent with the nonsynonymous SFS. We then use these results to parameterize simulations revisiting the question of whether the out-of-Africa bottleneck has led to differences in genetic load between African and non-African human populations. Altogether, our analysis represents a comprehensive exploration of dominance in humans, with numerous implications for understanding the relevance of recessive deleterious variation in humans and other species.

## Results

### Inference of dominance assuming a single dominance coefficient

We first tested whether models with a single dominance coefficient (*h*) can fit the SFS for nonsynonymous mutations in the human genome. To do this, we followed the same approach as Kim et al. [27]. We generated the SFS for synonymous and nonsynonymous mutations from 432 individuals with European ancestry from the 1000 Genomes Project [38], then used the synonymous SFS to infer a demographic model consisting of a bottleneck followed by recent exponential growth (see **Methods** and **S1–S2 Tables**). We conditioned on this demographic model for all subsequent inferences. Note that in Fit∂a∂i, fitness is parameterized such that the mutant homozygote has fitness 1-2*s* and the heterozygote has fitness 1-2*sh*, whereas in many other contexts, the homozygote fitness is assumed to be 1-*s* (see below). Unless otherwise indicated, the remainder of the manuscript uses the Fit∂a∂i parameterization.

We began by assuming that the DFE follows a gamma distribution, since previous work has suggested that a gamma distribution is a reasonable functional form for the DFE [26–29]. When assuming that all mutations are additive (*h* = 0.5), the best-fitting DFE has a log-likelihood (LL) of -1450.58 (**Fig 1A** and **S3 Table**) and the shape ($\alpha$) and scale ($\beta$) parameters inferred assuming *h* = 0.5 are similar to those inferred from previous studies for humans [26,27,39]. We then re-inferred the parameters of the gamma distribution for the DFE when assuming different values of *h* (**Fig 1** and **S3 Table**). We found that the log-likelihood greatly

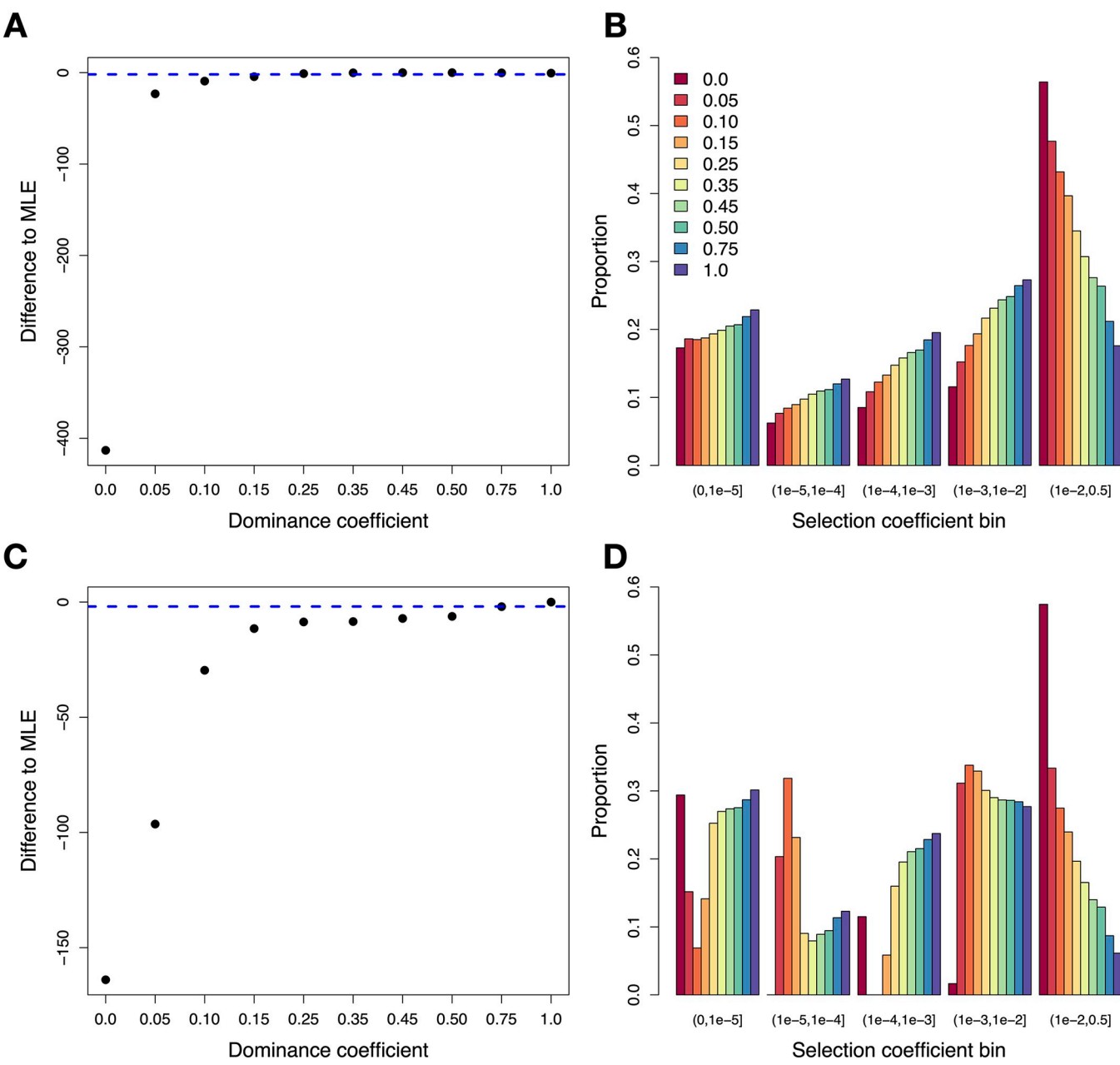

**Fig 1. Results when fitting a gamma (top panels) and discrete (bottom panels) DFE model under different dominance coefficients.** (A) Profile log-likelihood for the gamma. The *y*-axis shows the difference in log-likelihood relative to the best-fitting model (*h* = 0.50, LL = -1450.58) and the dashed blue line depicts 1.92 log-likelihood units below the top model. (B) The proportion of mutations for each selection coefficient bin under the gamma model when assuming different dominance coefficients. Note that the gamma model assumes a continuous distribution but results are shown here as discrete bins to facilitate visualization. (C) Profile log-likelihood for the discrete DFE model. The *y*-axis shows the difference in log-likelihood relative to the best-fitting model (*h* = 1.0, LL = -1446.76). (D) The proportion of mutations for each selection coefficient bin under the discrete DFE model when assuming different dominance coefficients.

decreased for models with $h<0.15$, suggesting that highly recessive models do not fit the data well (**S1 Fig**), in agreement with previous work [30]. However, other models where all nonsynonymous mutations are partially recessive ($h = 0.35$, $\Delta\text{LL}_{\text{additive}} = 0.2$) or dominant ($h = 0.75$, $\Delta\text{LL}_{\text{additive}} = 0.25$) are within 1.92 LL units of the additive model, suggesting that these models also provide a reasonable fit to the data. As the assumed value of $h$ becomes more recessive, the

shape parameter of the DFE tends to decrease, while the scale increases (S3 Table). Consequently, the average selection coefficient ($E[s]$) becomes more deleterious as $h$ decreases (S3 Table), as expected given that the overall product of $h*s$ should remain constant. Indeed, we find that $h*s$ is generally around ~0.0034 for gamma models with good fit, though increases to ~0.0045 for highly recessive models where the fit to the SFS was poor (S3 Table and S1 Fig).

Next, we examined the fit of a discrete DFE including five bins of $s$ for new mutations: neutral ($0 < |s| \leq 10^{-5}$), nearly neutral ($10^{-5} < |s| \leq 10^{-4}$), weakly deleterious ($10^{-4} < |s| \leq 10^{-3}$), moderately deleterious ($10^{-3} < |s| \leq 10^{-2}$), and strongly deleterious ($10^{-2} < |s| \leq 0.5$). The discrete DFE quantifies the proportion of mutations in each bin, where $s$ is uniformly distributed within each bin. The advantage of using a discrete DFE is that it does not assume a unimodal distribution for $s$. Further, as shown below, this DFE more easily allows for mutations with different values of $s$ to have different values of $h$. When assuming that all mutations are additive, we found that the fit of the discrete DFE to the SFS is slightly worse than the fit of the gamma DFE (LL = -1452.97, $\Delta$AIC to gamma = 8.8; S4 Table), consistent with what was observed previously [27]. We then re-inferred the parameters of the discrete DFE when assuming different values of $h$ (Fig 1C and 1D). Under the discrete DFE, we found that models with $h$ ranging from 0.15 to 1.0 fit the data well, with a maximum log-likelihood of -1446.76 when $h$ = 1.0 (Figs 1C and S2 and S4 Table). However, we observed a very poor fit for models where mutations were highly recessive ($h < 0.15$)(Figs 1C, 1D and S2). In these highly recessive models, we again observe that the proportion of mutations that are strongly deleterious increases substantially (Fig 1D), with nearly 60% of new mutations inferred to have $|s| > 10^{-2}$ when $h$ = 0.0. By contrast, when $h$ = 1.0, only ~6% of mutations are inferred to have $|s| > 10^{-2}$ (Fig 1D).

## Fitting models with multiple dominance coefficients

Our analyses above assumed a single dominance coefficient for all nonsynonymous mutations. However, such models are likely unrealistic, as $h$ is thought to vary among different classes of deleterious mutations [14,15,17,18,25]. To systematically explore the parameter space for recessive deleterious mutations in humans, we examined the fit of models where $h$ could differ for each bin of the discrete DFE. To focus on more biologically plausible models, we assumed that neutral mutations ($|s| < 10^{-5}$) are additive and that all the other mutations could have $h$ between 0 and 0.5. These constraints led to a total of 4096 $h$-$s$ models to be tested. For each model, we considered values of $h$ for each bin including 0.0, 0.05, 0.10, 0.15, 0.25, 0.35, 0.45, and 0.50. For a given combination of $h$ values, we then inferred the proportions of mutations in each bin of the discrete DFE that maximizes the log-likelihood. Given the challenges of identifying the 'true' model in this large parameter space, we validated our approach on simulated data, finding that the true model was within 2.54 LL units of the MLE (see Methods and S3 Fig). Based on this finding, we used a cut-off of 4.74 LL units (the asymptotic 95% confidence interval for a model with 4 free parameters) to designate models with good fit.

We found that these 4096 dominance models produced a wide range of DFE estimates, many exhibiting a very poor fit to the data (Fig 2, top row). After removing a total of 3312 models that were significantly different than the MLE (4.74 LL units lower than the best model's LL = -1451.68), 784 models remained with good fit to the nonsynonymous SFS (hereafter, "high LL models"). These models demonstrate that a range of $h$ values for each bin, ranging from additive to fully recessive, can yield a good fit to the nonsynonymous SFS (Fig 2, middle row). However, the data constrains some of the parameter space. For example, models with $h < 0.15$ for moderately deleterious mutations ($10^{-3} < |s| \leq 10^{-2}$) do not fit the data well, suggesting that such mutations are either additive or only partially recessive. Strongly deleterious mutations ($10^{-2} < |s| \leq 0.5$) can be more recessive, as models with $h$ = 0.05 fit the data (Fig 2).

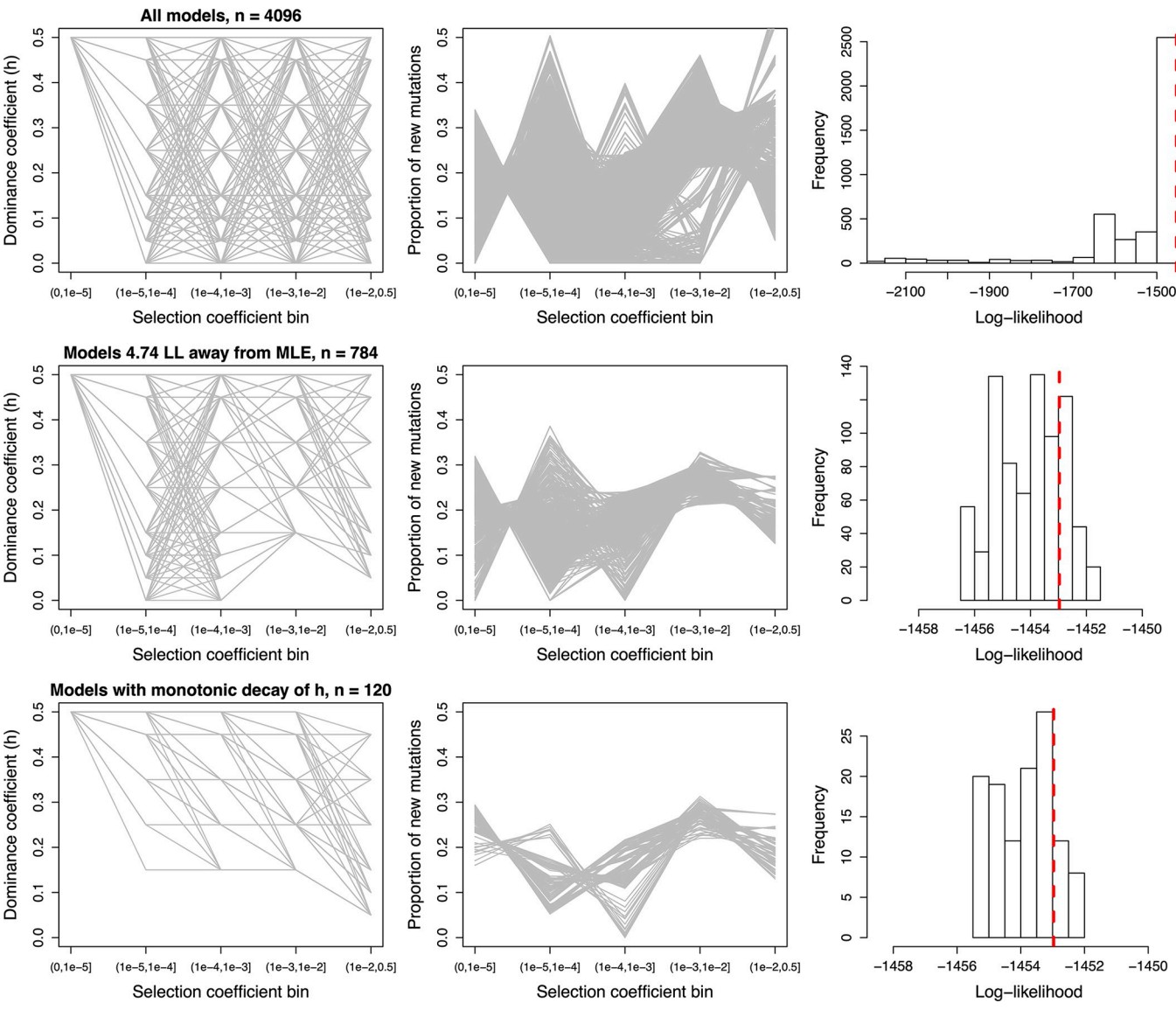

**Fig 2. Visualizing DFE and dominance parameters for models with varying *h* for different discrete bins of *s*.** Left column: Representation of the *h-s* relationship for each model. Each model is shown as a gray line. Middle column: The resulting DFE inferred assuming each *h-s* relationship. Each model is shown as a gray line. Right column: The distribution of log-likelihood values when fitting different *h-s* discrete models. Red dashed line corresponds to the log-likelihood under an additive discrete model with 4 parameters (-1452.97). Top: All *h-s* models considered. Middle: Only those *h-s* models having a log-likelihood<4.74 units from that of the MLE are shown. Bottom: Only those *h-s* models having a log-likelihood<4.74 units from that of the MLE and that have a monotonic relationship between *h* and *s* are shown.

Consistent with the data rejecting highly recessive models (*h*<0.05), we found that a highly recessive *h-s* relationship that was previously inferred for *Arabidopsis* by Huber et al. [25] also did not yield a good fit to human data (**S4 Fig**). This finding suggests that the parameters of the *h-s* relationship may greatly differ across species.

As another way of visualizing the impact of recessive mutations on model fit, we plotted the change in log-likelihood relative to a fully additive model while assuming progressively more recessive dominance coefficients for each selection coefficient bin of the DFE. In other words, we varied *h* for each bin of the DFE one-by-one while assuming *h* = 0.5 for all other bins and determined the effects on model fit. We found that model fit changed minimally while varying

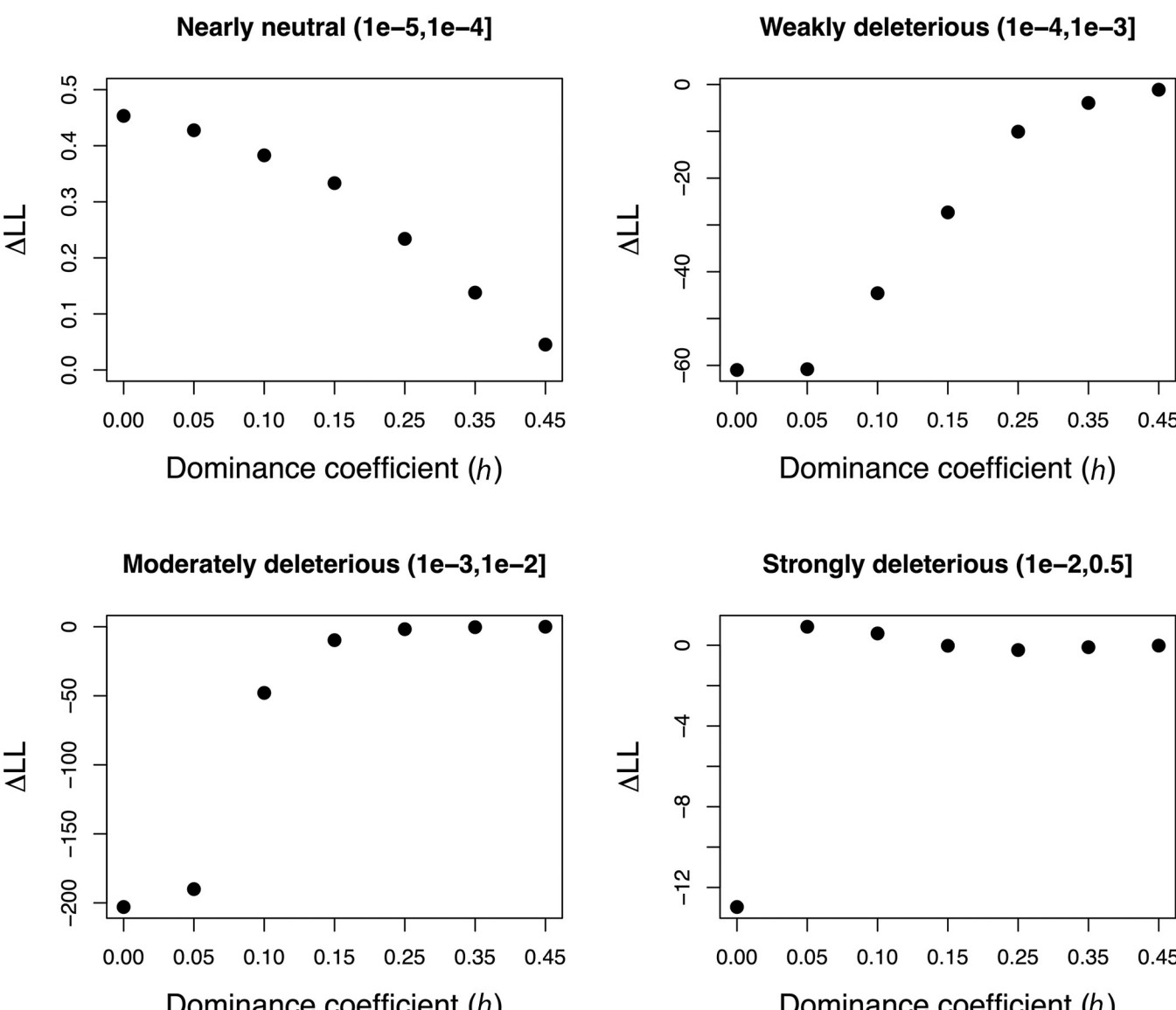

**Fig 3. Exploring the impact of changing _h_ for each selection coefficient bin under a discrete DFE model.** Each plot shows the change in log-likelihood (ΔLL) relative to a model where all bins are assumed to be additive (_h_ = 0.5; LL = -1452.97). In each case, the dominance coefficient for the specified bin of the DFE (shown in each panel) was changed to a more recessive value (shown on the _x_-axis) while holding all other bins to _h_ = 0.5. Note that the model fit changes minimally as _h_ becomes more recessive with the exception of making the weakly or moderately deleterious bins recessive. Strongly deleterious mutations show a complex pattern, where a model of _h_ = 0.05 results in a slight improvement in fit compared to the additive case while a fully recessive model (_h_ = 0) fits worse.

_h_ for the nearly neutral bin, whereas much greater changes in model fit were observed for other bins of _s_ (**Fig 3**). Specifically, models with _h_ = 0.0 for the weakly deleterious and moderately deleterious bins are ~60 and ~200 LL units worse than the fully additive model, respectively (**Fig 3**). This suggests that, although some models with fully recessive weakly deleterious mutations can fit the data (**Fig 2**), the constraint that all other bins are additive yields a poor fit (**Fig 3**). Finally, for the strongly deleterious bin, we observe a loss of ~13 LL units when _h_ = 0.0, though find that the fit is slightly improved when _h_ = 0.05 relative to the fully additive model (**Fig 3**). This result is consistent with our above finding that strongly deleterious mutations appear to have a lower bound of _h_ of 0.05 (**Fig 2**).

**Table 1. Summary of model averaging results.** Results are shown when considering all possible models ($n = 4096$), high LL models within 4.74 LL units of the MLE ($n = 784$), and high LL models with a monotonic decay in $h$ ($n = 120$). For each model, parameters of the discrete DFE (i.e. the proportions of mutations falling in each bin) and corresponding $h$ for each bin of the DFE are shown. Selection coefficient bins are defined as: neutral ($0 < |s| \leq 10^{-5}$), nearly neutral ($10^{-5} < |s| \leq 10^{-4}$), weakly deleterious ($10^{-4} < |s| \leq 10^{-3}$) moderately deleterious ($10^{-3} < |s| \leq 10^{-2}$), and strongly deleterious ($10^{-2} < |s| \leq 0.5$). Note that models were constrained to enforce additivity ($h = 0.5$) for neutral mutations.

| | Neutral | Nearly neutral | Weakly deleterious | Moderately deleterious | Strongly deleterious | Average $h$ |
|---|---|---|---|---|---|---|
| **All models** | | | | | | |
| *DFE* | 0.229 | 0.125 | 0.117 | 0.281 | 0.248 | |
| *h* | 0.5 | 0.231 | 0.226 | 0.227 | 0.232 | 0.291 |
| **High LL** | | | | | | |
| *DFE* | 0.220 | 0.139 | 0.177 | 0.277 | 0.187 | |
| *h* | 0.5 | 0.233 | 0.387 | 0.339 | 0.240 | 0.350 |
| **Monotonic decay** | | | | | | |
| *DFE* | 0.245 | 0.144 | 0.132 | 0.266 | 0.214 | |
| *h* | 0.5 | 0.451 | 0.377 | 0.300 | 0.189 | 0.358 |

## Testing models with an *h-s* relationship

Because experimental and molecular population genetics work has previously suggested that more deleterious mutations tend to be more recessive [14,17,18,25], we next restricted all the high LL models ($n = 784$) to those with a monotonic decay in their dominance coefficients from the neutral to the strongly deleterious mutation classes. A total of 120 high LL models with a monotonic decay remained, with log-likelihoods ranging from -1452.02 to -1453.50 (**Fig 2**, bottom row). Under these conditions, nearly neutral, weakly deleterious, and moderately deleterious mutations have a lower bound on the dominance coefficient of $h = 0.15$, while strongly deleterious mutations have a lower bound of $h = 0.05$. Across these different models, the inferred discrete DFE proportions remain relatively consistent for some selection coefficient bins, though are variable for others. For instance, the inferred proportions for the moderately deleterious bin remained within 0.22–0.31, whereas the strongly deleterious bin varied more widely between 0.13–0.27 (**Fig 2**).

To obtain estimates of $h$ and the DFE while accounting for model uncertainty, we next conducted model averaging using the $\Delta$AICs as weights of contributions to the parameter estimation (see **Methods**). When averaging across all 4096 models, we estimate an overall $h$ of 0.29, including an estimate of $h = 0.23$ for strongly deleterious mutations (**Table 1**). Model averages when using high LL models ($n = 784$) and monotonic high LL models ($n = 120$) both suggest an overall $h$ of ~0.35 along with DFEs that are highly similar (**Table 1**). However, the estimated $h$ values for each bin of the DFE are somewhat variable, particularly for the nearly neutral bin (**Table 1**). Finally, although such models with a monotonic decay are expected based on previous work [14,17,18,25], we find that these monotonic decay models on average do not fit any better than high LL models without a monotonic decay (mean LL for monotonic decay = -1453.64, mean LL for high LL without monotonic decay = -1453.46; **Fig 2**). Thus, our analysis does not provide any statistical support for or against the presence of an *h-s* relationship.

## Simulating deleterious variation in human populations under a range of dominance models

Our results suggest a range of dominance models can fit human genetic variation data, though with support for an overall mean $h$ for nonsynonymous mutations of ~0.35 and evidence for the possibility of an *h-s* relationship. In light of these findings, our next aim was to explore how these dominance parameters may influence models of deleterious variation and genetic

**Table 2. Summary of DFE & dominance models used for simulations.** For each model, parameters of the discrete DFE and corresponding *h* for each bin of mutations are shown. Selection coefficient bins are defined for SLiM simulations as: neutral ($0<|s|\leq2\times10^{-5}$), nearly neutral ($2\times10^{-5}<|s|\leq2\times10^{-4}$), weakly deleterious ($2\times10^{-4}<|s|\leq2\times10^{-3}$) moderately deleterious ($2\times10^{-3}<|s|\leq2\times10^{-2}$), and strongly deleterious ($2\times10^{-2}<|s|\leq1$).

| | Neutral | Nearly neutral | Weakly deleterious | Moderately deleterious | Strongly deleterious | Average *h* |
|---|---|---|---|---|---|---|
| **Strongly recessive** | | | | | | |
| **DFE** | 0.201 | 0.222 | 0.018 | 0.286 | 0.274 | |
| ***h*** | 0.5 | 0.45 | 0.25 | 0.15 | 0.05 | 0.26 |
| **Moderately recessive** | | | | | | |
| **DFE** | 0.244 | 0.152 | 0.125 | 0.259 | 0.22 | |
| ***h*** | 0.5 | 0.5 | 0.45 | 0.25 | 0.1 | 0.34 |
| **Weakly recessive** | | | | | | |
| **DFE** | 0.259 | 0.125 | 0.176 | 0.254 | 0.186 | |
| ***h*** | 0.5 | 0.5 | 0.5 | 0.35 | 0.15 | 0.40 |

load across human populations. Specifically, we sought to revisit the question of how the out-of-Africa bottleneck may have impacted the relative burden of deleterious variation in African and non-African human populations. As previous studies have demonstrated that dominance is a key component influencing potential differences in deleterious burden [6–10], reevaluating these patterns with dominance models that are fit to human genetic variation data may provide further clarity on this topic.

To explore how the relative burden of deleterious variation in African and non-African human populations behaves under our estimated dominance and selection parameters, we ran forward-in-time simulations of deleterious genetic variation using SLiM [40–42] under a human demographic model [43] employing a range of dominance models as suggested by our results. These models include a "Weakly Recessive" model with average $h = 0.40$ and $h = 0.15$ for strongly deleterious mutations, a "Moderately Recessive" model with average $h = 0.34$ and $h = 0.10$ for strongly deleterious mutations, and a "Strongly Recessive" model with average $h = 0.26$ and $h = 0.05$ for strongly deleterious mutations (**Table 2**). We chose these models to encompass the spectrum of inferred dominance parameters from our results while also exploring plausibly low dominance coefficients for strongly deleterious mutations, given the widespread evidence for strongly deleterious mutations being highly recessive [14,15,17,25,44,45]. Importantly, the above models were selected from the broader set of 120 high LL models with a monotonic decay (**Fig 2 and Table 2**); thus, these models all exhibit similarly good fit to the human nonsynonymous SFS. Finally, note that SLiM parameterizes the homozygote fitness as 1-*s* and heterozygote fitness 1-*sh*, thus selection parameters inferred from Fit∂a∂i were converted accordingly to SLiM parameters (see **Methods and Table 2**).

For these simulations, we modelled coding regions for 22 autosomes, yielding a total sequence length of ~30Mb and deleterious mutation rate of $U = 0.63$ (see **Methods**). Under each dominance model, we outputted the predicted genetic load at the conclusion of the simulation, which measures the reduction in mean population fitness due to segregating and fixed deleterious mutations [5,35]. Additionally, we also outputted the predicted inbreeding load under each model, which measures the potential severity of inbreeding depression (i.e., the quantity of recessive deleterious variation that is concealed as heterozygotes) in a population [4,35,46]. The inbreeding load (often referred to as the 'number of lethal equivalents' or *B*) therefore provides a complementary perspective on the burden of deleterious variation. Moreover, several empirical inbreeding load estimates exist for humans, suggesting a range for *B* between ~0.7–2.5 [46,47]. Thus, comparing these empirical estimates to those predicted by each dominance model can serve as an additional source of evidence to validate dominance and selection parameters.

Our simulation results suggest that the predicted patterns of load depend somewhat on the dominance model employed. Specifically, in the Weakly and Moderately Recessive models, we observe a slight increase in genetic load in the non-African population (1.9% and 1.1% increase, respectively), whereas in the Strongly Recessive model, we observe a 1.5% decrease in genetic load in non-African populations (**Fig 4A**). However, all models exhibit a slight 'purging' of the inbreeding load in the non-African population due to the out-of-Africa bottleneck (3.8% for the Weakly Recessive model, 2.8% for the Moderately recessive model, and 3.3% for the Strongly Recessive model; **Fig 4B**). Moreover, the total inbreeding load predicted varies by model, with $B = {\sim}0.55$ predicted for the Weakly Recessive model, $B = {\sim}0.92$ predicted for the Moderately Recessive model, and $B = {\sim}2.1$ predicted for the Strongly Recessive model (**Fig 4B**). Notably, this range of predicted inbreeding loads is in good agreement with the range of empirical estimates in humans of ~0.7–2.5 [46,47]. Finally, the counts of derived nonsynonymous alleles generally mirrored the patterns of genetic load, with a 0.9% increase in the non-African population under the Weakly Recessive model, a 1.0% increase under the Moderately Recessive model, and a 1.2% decrease under the Strongly Recessive model (**Fig 4C**).

## The impact of variable $h$ for a given $s$

All of the dominance models considered so far assume a single $h$ for a given $s$. However, it is well-appreciated from studies of Mendelian disorders that loci with similar phenotypic effects can range from dominant to fully recessive [48]. Moreover, experimental studies in organisms such as yeast and *Drosophila* also support the possibility that $h$ may vary for a given $s$ [14,49]. To explore the impact of variable $h$ in the context of our results, we ran simulations under the Strongly Recessive model from above; however, we allowed $h$ for each bin of $s$ to be drawn from a uniform distribution bounded by 0 and $2*\bar{h}$. For instance, in the case of the strongly deleterious bin where $\bar{h} = 0.05$, $h$ could range from 0 to 0.1.

With this model, we first examined the impact of variable $h$ on the predicted nonsynonymous SFS (**S5 Fig**). We observed no noticeable differences between the predicted nonsynonymous SFS when comparing models with fixed and varying $h$, and both models closely matched the observed SFS from the 1000 Genomes Project data (**S5 Fig**). Thus, this result demonstrates that modest variability in $h$ for a given $s$ might not impact overall patterns of genetic variation, implying that our inference approach can infer the average $h$ for each bin of $s$ even when variability is present.

We next employed this Strongly Recessive model with variable $h$ to explore the potential impact of variable $h$ on genetic load and inbreeding load. Here, we find that our previous qualitative finding for the Strongly Recessive model (**Fig 4**) of diminished genetic load and inbreeding load in non-African populations was recapitulated under a model with variable $h$ (**S6 Fig**). However, the magnitude of genetic load was shifted slightly down, whereas the magnitude of the inbreeding load was greatly shifted up (mean $B = {\sim}3.5$ across populations; **S6 Fig**). This may be due to the increased number of mutations that have the potential to be highly recessive ($h<0.05$) in this model, given that all deleterious mutation classes were assumed to have a lower bound of $h = 0.0$. Overall, these results demonstrate that variability in $h$ may have important impacts on genetic load and inbreeding depression, suggesting that future studies should aim to better parameterize variability in $h$ for deleterious mutations in humans and other species.

## Discussion

Here, we have investigated the fit of dominance and selection models for nonsynonymous mutations in humans and explored implications for the relative burden of deleterious variation

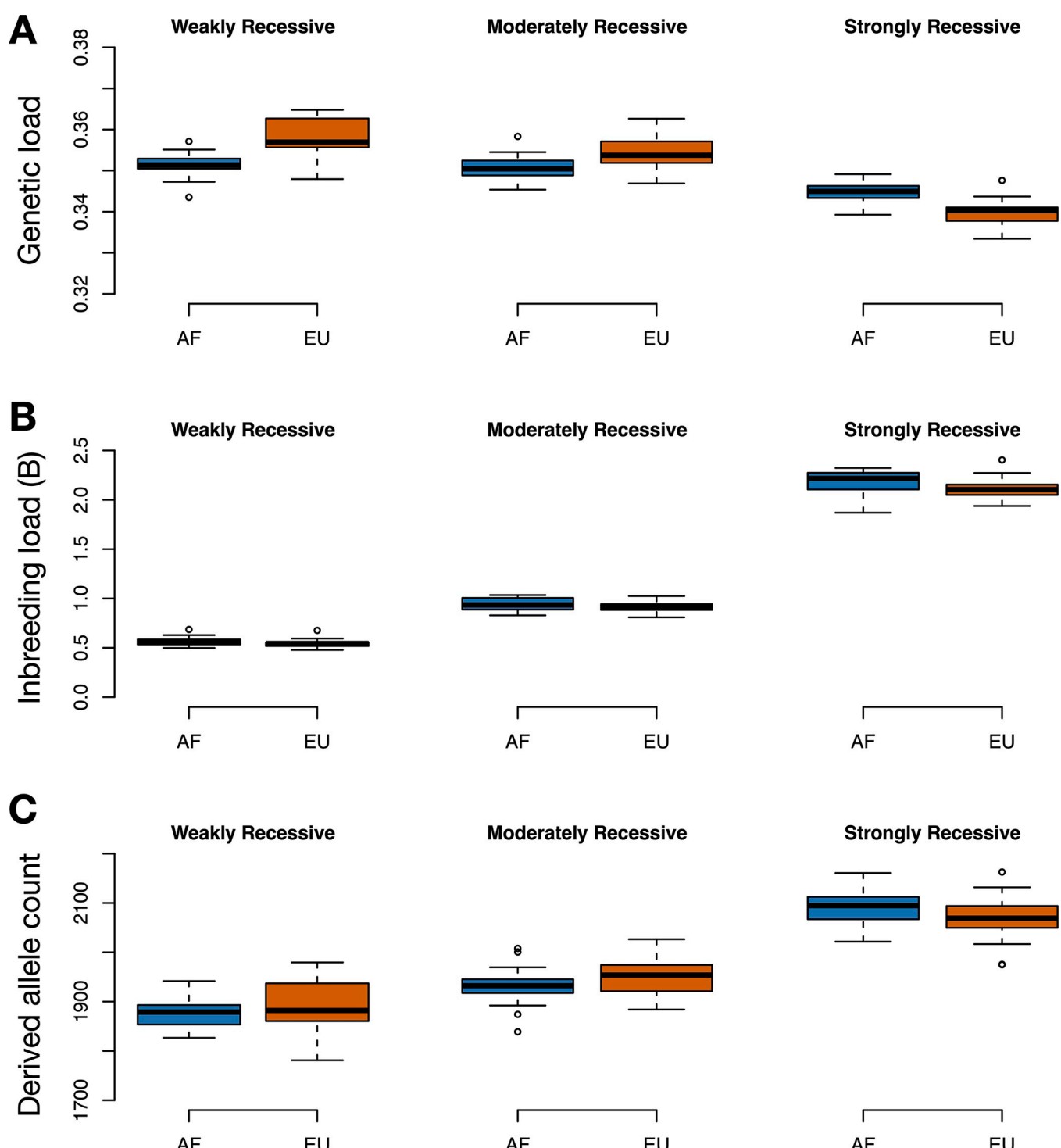

**Fig 4. Simulation results comparing predicted genetic load, inbreeding load, and derived allele count for African (AF) and European (EU) populations under four different DFE and dominance models.** (A) Predicted genetic load in African and European populations. (B) Predicted inbreeding load in African and European populations. Note that empirical estimate of *B* in humans range from ~0.7–2.5 [46,47]. (C) Predicted derived deleterious allele count in African and European populations. Results are shown as boxplots summarizing output from 25 simulation replicates under each DFE and dominance model. See **Table 2** for details on each DFE and dominance model.

in African and non-African human populations. Our results demonstrate that a wide range of dominance models are consistent with patterns of nonsynonymous genetic variation in humans, including models with a strong *h-s* relationship. Although our analysis is unable to fully overcome issues of identifiability for *h* and *s* parameters, some general conclusions can be drawn. First, we demonstrate that highly recessive models with a single dominance coefficient ($h < 0.15$) cannot yield a good fit to the nonsynonymous SFS in humans (**Figs 1, S1 and S2**). Next, we find that many models with an *h-s* relationship fit the data (**Fig 2**), as predicted by previous theoretical and empirical work [14,17,19,25,50]. Additionally, through model averaging, we estimate an average *h* for nonsynonymous mutations on the order of ~0.35, though with much greater uncertainty for *h* values of individual selection coefficient bins (**Fig 2 and Table 1**). Notably, we find that the *Arabidopsis h-s* relationship parameters estimated by Huber et al. [25] do not fit human data (**S4 Fig**), suggesting that dominance parameters are likely to vary across species. Finally, we also find that that models with $h = 0.05$ for strongly deleterious mutations provide a small improvement in fit to the SFS (**Fig 3**), consistent with previous work demonstrating that such mutations are likely to be highly recessive [14,15,17,25,44,45]. For instance, the impacts of recessive lethal mutations in humans are well documented [44,51,52], though evidence from *Drosophila* suggests that such lethal mutations may in fact have a small fitness consequence in the heterozygote state [17], consistent with our findings (**Figs 2 and 3**)

Dominance is an essential determinant of the influence of demography on patterns of deleterious variation [2,3,5,6]. In humans, previous work has found that dominance plays a key role in determining the impact of the out-of-Africa bottleneck on relative patterns of deleterious variation and genetic load in African and non-African populations [6–10]. In the absence of estimates of dominance parameters in humans, previous studies have typically assumed the extreme cases of additive and fully recessive models [9–11] or used *ad hoc* dominance parameters [7]. Our simulation results under a range of plausible dominance models fit to data further demonstrate the subtle influence of dominance on the relative burden of genetic load in African and non-African populations. Specifically, we find that genetic load may be slightly elevated in non-African populations when deleterious mutations are weakly or moderately recessive, whereas genetic load in non-African populations may in fact be slightly diminished if deleterious mutations are more strongly recessive (**Fig 4**). This diminished genetic load in non-African populations in the Strongly Recessive model is driven by a slight purging of the inbreeding load during the out-of-Africa bottleneck (**Fig 4**), a process that is known to be most efficient for highly recessive strongly deleterious mutations [1–3]. Finally, these simulation results also help to further validate our dominance models, demonstrating that selection and dominance models estimated from genomic variation datasets predict inbreeding loads that are broadly consistent with empirical measures. Specifically, these models predict an inbreeding load between ~0.5–2.1, a range that is strikingly similar to that suggested by empirical studies of ~0.7–2.5 [46,47]. This result helps bridge the gap between molecular studies of dominance and selection parameters and more direct measures of fitness in humans, suggesting that results from these very different approaches can be reconciled. Moreover, this result also suggests that nonsynonymous mutations alone can account for much of the inbreeding depression observed in humans. Notably, our results with variable *h* for a given *s* predict a much higher inbreeding load compared to models with a fixed *h* for each bin of *s*, suggesting that variability in *h* may be an important element influencing the magnitude of inbreeding depression.

Our work also has implications for studies of deleterious variation in non-human taxa. In particular, there has been a great deal of recent interest in modelling the impact of recessive deleterious variation on extinction risk in small and isolated populations [3,35,53–55]. Our

analysis can help guide such studies by informing the parameterization of dominance and selection models, as many of these studies are focused on endangered species of mammals that may have similar population genetic parameters to humans [35]. For instance, it has been previously suggested that the historical population size of a species may greatly influence risk of extinction due to inbreeding depression, particularly in cases where strongly deleterious alleles ($0.01 < |s| \leq 1$) are highly recessive ($h < 0.05$) [2–4]. The results of our simulation analysis indicate that models with fully recessive strongly deleterious mutations are not compatible with empirical estimates of the inbreeding load in humans (**Fig 4**), suggesting that this relationship between demography and deleterious variation may be somewhat dampened, at least in humans. Instead, we find that an *h* for strongly deleterious mutations on the order of ~0.05–0.15 could better explain empirical estimates of the inbreeding load (**Fig 4**). However, we note that a major limitation of our study is that we are unable to obtain fine-scale estimates of selection and dominance parameters for strongly deleterious mutations, which as defined here encompass a wide range of |*s*| from 0.01 to 1. This limitation is due to SFS-based methods being underpowered for estimating the strongly deleterious tail of the DFE [52], due to the fact that such mutations tend not to be segregating in genetic variation datasets [56–58]. Moreover, our diffusion-based approach may also be limited in inferring dominance parameters for strongly deleterious mutations given that the diffusion approximation breaks down under strong selection [59]. Given these considerations, our finding of a lower bound of $h = 0.05$ for strongly deleterious mutations should be interpreted with some caution. Future work should focus on further refining selection and dominance parameters for strongly deleterious mutations.

Our study also remains limited by an inability to statistically favor one dominance model over another. For example, an additive model continues to fit the SFS very well, exhibiting similar fit to more complex recessive models (**Figs 1 and 2 and S1 and S2 Tables**). To overcome this, we use a model averaging approach (**Table 1**), which highlights some general trends, though does not necessarily indicate that complex models with recessive mutations are an improvement over an additive model. Nevertheless, the plausibility of these more complex models with highly recessive strongly deleterious mutations are supported by previous theoretical and experimental work on dominance in non-human taxa [14,17,19,25,50] as well as the broad literature on recessive disease and inbreeding depression in humans [12,44,46,47,60–64]. Although we remain unable to determine an optimal dominance model in humans, our work provides a range of useful models that can be employed in future analyses [65]. Ultimately, any approach for inferring dominance and selection parameters from allele frequency data will have limited power to disentangle *h* and *s* in outbreeding species [24,25]. Indeed, we find that models with a good fit to the data have an average *h*\**s* in the range of 0.003–0.025 (**S7 Fig**). However, not all DFE and dominance models with an inferred average *h*\**s* in this range fit the data well, as there are many combinations of *s* and *h* values for different bins of the DFE that yield similar average *h*\**s* values. Furthermore, our aim of separately considering *h* and *s* parameters in this study has practical benefits in that it can yield parameter combinations that can be used in simulation studies (**Fig 4**).

To improve on our findings and obtain more precise estimates of dominance in humans, some recent work has suggested that there may be information on dominance from patterns of linkage disequilibrium [66,67] or from patterns of transmission in pedigrees [68]. Alternatively, leveraging genomic datasets from populations with varying levels of inbreeding in humans or other mammals, as done by Huber et al. [25] in *Arabidopsis*, may also offer a fruitful avenue for research. Future studies should continue to explore these and other avenues for inferring dominance parameters to better inform our understanding of the evolutionary significance of dominance.

## Materials and methods

### Data

We downloaded SNP genotype data for 432 individuals with European ancestry from the 1000 Genomes Project phase 3 release [38] from http://ftp.1000genomes.ebi.ac.uk/vol1/ftp/release/20130502/. This European sample includes individuals from five different European populations (Utah residents with Northern and Western European ancestry, British in England and Scotland, Finnish, Iberian populations in Spain, and Toscani in Italy), among which there is pronounced population structure [38] that could potentially impact DFE inference. However, previous simulation analysis by Kim et al. [27] demonstrated that such cryptic population structure does not impact the performance of Fit∂a∂i, so long as the demographic model can fit the synonymous SFS. Additionally, this paper also showed that the DFE inferred using the European sample from the 1000 Genomes Project was similar to the DFEs inferred from a more homogenous sample from Denmark as well as from the NHLBIO GO Exome Sequencing Project (ESP) data, which consists of a heterogenous sample of individuals with European ancestry [27].

We followed filtering steps as described in Kim et al. [27]. Specifically, only unrelated individuals were used and only sites from exome-targeted sequencing that passed the strict mask filter criteria (as defined in [38]) were kept. We then extracted the synonymous and nonsynonymous sites based on the 1000 Genomes Project-filtered annotations. The folded site frequency spectra were computed by tabulating the observed counts of the minor allele frequencies for synonymous and nonsynonymous variants separately. The synonymous and nonsynonymous site frequency spectra were then used for the inference of demography and DFE, respectively. We also computed the length of synonymous ($L_S$) and the length of nonsynonymous sites ($L_{NS}$) sites which were used for the estimation of population genetic parameters (see below; S2 Table). Site frequency spectra for synonymous and nonsynonymous variants are presented in S5 and S6 Tables, respectively.

### Demographic parameters

Because demographic history distorts allele frequencies in similar ways to selection, we used the synonymous SFS to tease apart the effect of demography and selection on the SFS. Here, we assume that the synonymous SFS is neutral and that there is no linkage between synonymous and nonsynonymous sites, two assumptions that are likely violated in reality. However, Kim et al. [27] have previously shown that Fit∂a∂i is able to infer the true DFE parameters despite the presence of unmodeled linkage, and Martinez i Zurita et al. [69] have recently shown that small amounts of selection on synonymous variants does not impact DFE inference of nonsynonymous mutations.

Based on the synonymous SFS, we inferred the parameters in a three-epoch demographic model (out-of-Africa European demographic model with the occurrence of a bottleneck, a recovery period, followed by a recent exponential population growth). We computed changes in population size relative to the ancestral population size ($N_{anc}$) by storing the maximum-likelihood estimates (MLEs) of demography and population sizes after 30 iterations (MLEs are shown in S1 Table). The mutation rate at synonymous sites ($\theta_S$) was then estimated as the scaling factor difference between the optimized SFS and the empirical data using the function *dadi.Inference.optimal_sfs_scaling* with ∂a∂i [70]. The ancestral population size $N_{anc}$ was obtained from the inferred scaled mutation rate at synonymous sites ($\theta_S$) using the formula: $\theta_S = 4N_{anc} \times \mu \times L_S$, where $L_S$ is the is total count of synonymous sites, and $\mu$ is the per-base-pair mutation rate. We assumed a mutation rate of $1.5 \times 10^{-8}$ mutation per site per generation [71].

To compute the nonsynonymous scaled mutation rate ($\theta_{NS}$) we assumed a 2.31 ratio between $L_{NS}$ and $L_S$ [39]. All computed population genetic parameters are described in **S2 Table**.

## DFE inference under a gamma model with varying *h*

To infer selection and dominance parameters while accounting for demography, we assumed the estimated demographic parameters obtained from the synonymous SFS (**S1 Table**), and used the nonsynonymous SFS to infer the DFE with Fit$\partial$a$\partial$i using a Poisson likelihood function as described in [26,27,39]. In Fit$\partial$a$\partial$i, fitness is parameterized such that the mutant homozygote has fitness 1-2*s* and the heterozygote has fitness 1-2*sh*. Thus, selection coefficients for deleterious mutations are positive and range from 0 to 0.5, where 0.5 is considered a lethal mutation with fitness of 0.

Our initial aim was to fit a gamma distributed DFE to the nonsynonymous SFS under varying *h* values including 0.0, 0.05, 0.10, 0.15, 0.25, 0.35, 0.45, 0.50, 0.75, and 1.0. The gamma distribution has two parameters (shape and scale, often denoted $\alpha$ and $\beta$) and has previously been shown to provide a reasonable distribution for the DFE [26–29]. To determine the expected SFS for nonsynonymous mutations under varying values of *h*, we used the equation in Williamson et al. [23] implemented within $\partial$a$\partial$i in the function *phi_1D* to compute the quasi-stationary distribution of allele frequency for a given site under the assumptions of the Wright-Fisher model (random mating, constant population size, non-overlapping generations):

$$f(q;\ \gamma, h) = \frac{e^{4\gamma h q + 2\gamma(1-2h)q^2}}{q(1-q)} \frac{\int_q^1 e^{-4\gamma h \varepsilon - 2\gamma(1-2h)\varepsilon^2} d\varepsilon}{\int_0^1 e^{-4\gamma h \varepsilon - 2\gamma(1-2h)\varepsilon^2} d\varepsilon},$$

where *q* is the frequency of the derived nucleotide in the population, $\gamma$ is the scaled selection coefficient ($s \times 2N_{anc}$), and *h* is the dominance coefficient. We then used the *Integration.one_-pop* function to update *f(q; $\gamma$, h)* to become the transient distribution of allele frequencies after the population changed size. By assuming that sites are independent, we can expand the above formula to multiple sites by simply assuming that the allele frequency of a given mutant allele is a random draw from the above distribution.

Each entry of the SFS, $x_i$, is the count of the number of variants at which the derived nucleotide present *i* times in a sample size of *n* individuals, for *i* 1, 2,..., *n*−1. Then the expected value for each entry of the SFS vector is $\theta F(n, i; \gamma, h)$ [23]:

$$F(n, i;\ \gamma, h) = \int_0^1 \binom{n}{i} q^i (1-q)^{n-i} f(q;\ \gamma, h) dq,$$

where $\theta$ is the per generation mutation rate of the sampled region. If mutations enter a population in each generation following a Poisson distribution [22], then each entry of the SFS ($x_i$) is expected to be Poisson distributed [22]. Given the full probability distribution of each entry of the SFS, the model parameters can be estimated in a maximum-likelihood framework within Fit$\partial$a$\partial$i.

We used this approach as implemented in Fit$\partial$a$\partial$i to compute the expected nonsynonymous SFS over a grid of 1000 log-spaced values of the population-scaled selection coefficient $\gamma$ ($s \times 2N_{anc}$). To maintain computational feasibility and avoid issues of numerical instability with the diffusion approximation due to large *s* [72], we restricted the range of *s* to $10^{-5}$ to 0.25, thus assuming that mutations with |*s*|>0.25 were not segregating in the 1000 Genomes Project dataset. We then inferred the shape and scale parameters of the gamma distribution from 25 runs of Fit$\partial$a$\partial$i by integrating over the grid of expected SFS and fitting model output to the empirical nonsynonymous SFS. Parameters with the highest Poisson log-likelihood

were chosen. For plotting, the scale parameter was divided by $2N_{anc}$ to no longer be scaled by the ancestral population size. Details of parameters used for inference and rescaling are provided in **S2 Table**.

### Inference under a discrete DFE model with varying *h*

In addition to testing a gamma distribution for the DFE, we also examined the fit of the discrete DFE under varying dominance coefficients. The discrete DFE can be seen as a mixture of uniform distributions consisting of five bins defined based on a range of selection coefficients. The bins were defined as: neutral ($0 < |s| \leq 10^{-5}$), nearly neutral ($10^{-5} < |s| \leq 10^{-4}$), weakly deleterious ($10^{-4} < |s| \leq 10^{-3}$), moderately deleterious ($10^{-3} < |s| \leq 10^{-2}$), and strongly deleterious ($10^{-2} < |s| \leq 0.5$). The proportions of new mutations in each bin were the parameters inferred using Fit∂a∂i. Note that, even though there are 5 bins of the DFE, the constraint that all 5 bins must sum to 1 means that only 4 parameters are actually inferred from the data. To estimate these parameters, we used the same approach as outlined above for the gamma distribution.

While the gamma and discrete DFEs appear to have different inferred mean $h*s$ values when assuming $h = 0.5$ (gamma: 0.0034 discrete: 0.0173; **S3 Table** and **S7 Fig**), this is due to how strongly deleterious mutations influence the mean $h*s$ in the two models. When only considering mutations that are not strongly deleterious ($|s| < 0.01$), the mean $h*s$ values inferred from the two DFEs are much closer to each other (gamma: 0.0007, discrete: 0.00098). Thus, both the gamma and discrete DFEs provide similar conclusions in the part of the parameter space with the greatest statistical power.

### Inference under a discrete DFE with multiple dominance coefficients

To better explore the fit of models with recessive mutations, we tested models where each bin of the discrete DFE could have its own value of *h* including 0.0, 0.05, 0.10, 0.15, 0.25, 0.35, 0.45, 0.50. To constrain the set of possible models, we assumed that neutral mutations ($0 < |s| \leq 10^{-5}$) were additive ($h = 0.5$) and tested out all combinations of *h* for the four other bins of the discrete DFE. This resulted in a total of 4096 (= $8^4$, where 8 is the number of *h* values tested and 4 is the number of bins of the DFE) models being fit to the nonsynonymous SFS.

To fit these models, we used the same approach outlined above including estimating a demographic model using the synonymous SFS and then estimating the parameters of the discrete DFE using the nonsynonymous SFS. For each bin of the discrete DFE, we used the precomputed expected SFS under the assumed value of *h* for the given range of *s*. For instance, in the case of a model with $h = 0.5$ for neutral mutations, $h = 0.35$ for nearly neutral mutations, $h = 0.25$ for weakly deleterious mutations, $h = 0.15$ for moderately deleterious mutations, and $h = 0.05$ for strongly deleterious mutations, we created a grid of expected nonsynonymous SFS using expectations for $h = 0.5$ for the *s* range from 0 to $10^{-5}$, $h = 0.35$ for the *s* range from $10^{-5}$ to $10^{-4}$, $h = 0.25$ for the *s* range from $10^{-4}$ to $10^{-3}$, $h = 0.15$ for the *s* range from $10^{-3}$ to $10^{-2}$, and $h = 0.05$ for the *s* range from 0.01 to 0.5. To maintain computational feasibility, we ran only 5 Fit∂a∂i runs for each of the 4096 dominance models and picked the parameters from the run with the highest Poisson log-likelihood. MLE parameter estimates and log-likelihoods for each of these models are provided in **S7 Table**.

Given the large parameter space of models being fit, we assessed using simulated data whether this inference approach would be able to identify the true model from the large set of 4096 models being fit. To do this, we simulated nonsynonymous sites across 22 autosomes (totaling ~30Mb of sequence) under the single population demographic model from Kim et al. [27] using SLiM v4.0.1 [40–42]. We assumed selection and dominance parameters from the Strongly Recessive model (**Table 2**) and all other parameters were the same as our main

simulation analysis described below. We then used the resulting nonsynonymous SFS as input for the inference approach described above.

One potential concern with parameterizing a discrete DFE in terms of a uniform distribution of *s* within each bin and a specific value of *h* for that bin is that different values of *h* will result in models having different *h\*s* values. As *h\*s* largely influences the SFS, the fit of different models could potentially be driven by changes in *h\*s* model space, rather than dominance, per se. To check this, we determined the location of each of the 4096 models in average *h\*s* model space. To do this, we multiplied the value of *h* for a particular bin of the discrete DFE by the expected value of *s* for that bin. Then we averaged this quantity across the 5 bins of the discrete DFE. There are gaps in the average *h\*s* model space that we considered (**S8 Fig**). However, this does not appear to influence our results as models with an average *h\*s* in the range of $5 \times 10^{-7}$–0.026 all have high log-likelihoods, indicating a satisfactory fit to the data (**S8 Fig**). Further, not all models within the average *h\*s* space $5 \times 10^{-7}$–0.026 fit the data well, reflecting the fact that models with different combinations of *h* and *s* can have similar average values in *h\*s* space.

## Computing AIC

To compare non-nested DFE models, such as the gamma and discrete models, and estimate parameters of interest using a model averaging approach (see below), we first transformed the log likelihood using the Akaike information criteria:

$$AIC = 2k - 2log(L)$$

where *k* is the number of estimated parameters (2 for the gamma DFE and 4 for the discrete DFE models) and *L* is the maximum likelihood of each model estimated. The preferred AIC model is the one with the minimum AIC value. These AIC values also were then used as weights of the model averaging (below).

## Model averaging with AIC

Because multiple *h-s* relationship models fit the data, we also used a model averaging approach to estimate the DFE and dominance parameters [73,74]. Let *x* refer to the parameter of interest (e.g. all entries in the discretized DFE and their respective dominance coefficients), $x_i$ refer to the MLE under model *i* that has an AIC value of $AIC_i$. In this approach, the parameter of interest, *x*, is estimated. Then, the model average estimate ($x_{avg}$) accounting for the contribution of each model is averaged using Akaike weights:

$$x_{avg} = \frac{\sum_i x_i e^{-1/2\Delta AIC_i}}{\sum_i e^{-1/2\Delta AIC_i}}.$$

The $\Delta AIC_i$ is obtained as:

$$\Delta AIC_i = AIC_i - min_i(AIC_i).$$

We computed model averages first considering all possible discrete models with varying *h* (**Fig 2**, top row), next considering only models that were 4.74 LL units away from the MLE (**Fig 2**, middle row), and finally considering only models that were 4.74 LL units away from the MLE and had a monotonic decay in *h* (**Fig 2**, bottom row).

## Simulation methods

We ran forward-in-time simulations under the Wright-Fisher model using SLiM v4.0.1 [40–42] employing DFE and dominance parameters inferred above to revisit the question of how

the out-of-Africa bottleneck has influenced the relative burden of deleterious variation in African and non-African human populations. We modelled deleterious alleles occurring as nonsynonymous mutations in coding sequence, with 22,500 genes on 22 autosomes each 1340bp in length, resulting in a total sequence length of 30.16 Mb [75]. Following Robinson et al. [31], we assumed a recombination rate of 0.001 crossovers per site per generation between genes, 0.5 between chromosomes, and no recombination within genes. The mutation rate for deleterious alleles was set to $1.5 \times 10^{-8} \times (2.31/3.31) = 1.05 \times 10^{-8}$ mutations per site per generation, where $1.5 \times 10^{-8}$ is the genome-wide mutation rate in humans [71] and 2.31/3.31 is the fraction of coding mutations that are assumed to be nonsynonymous [39]. We set the demographic parameters using the two-population model from Tennessen et al. [43], inferred from large samples of individuals of African and European descent. Specifically, this model assumes an ancestral population size of $N_e = 7,310$, followed by growth in Africa to $N_e = 14,474$, with the European population diverging 2,040 generations before present and experiencing a bottleneck at $N_e = 1,861$ for 1,120 generations, followed by exponential growth over the last 204 generations in both populations to $N_e = 424,000$ in Africa and $N_e = 512,000$ in Europe.

We employed three different DFE and dominance models in our simulations (**Table 2**), chosen to encompass the range of parameters observed in our monotonic h-s relationship results while also modelling strongly deleterious mutations as being highly recessive ($h \leq 0.15$) as informed by studies in other taxa [14,15,17,25]. These models include a "Weakly Recessive" model with average $h = 0.40$ and $h = 0.15$ for strongly deleterious mutations, a "Moderately Recessive" model with average $h = 0.34$ and $h = 0.10$ for strongly deleterious mutations, and a "Strongly Recessive" model with average $h = 0.26$ and $h = 0.05$ for strongly deleterious mutations (**Table 2**). Note that these models were selected from the broader set of 120 high LL models with a monotonic decay (**Fig 2** and **Table 1**) and therefore all exhibit a similar fit to the human nonsynonymous SFS. Finally, also note that, unlike Fit∂a∂i, fitness in SLiM parameterized such that the mutant homozygote has fitness 1-s and the heterozygote has fitness 1-sh. Thus, the selection coefficient bins for these SLiM simulations were defined as: neutral ($0 \leq |s| < 2 \times 10^{-5}$), nearly neutral ($2 \times 10^{-5} \leq |s| < 2 \times 10^{-4}$), weakly deleterious ($2 \times 10^{-4} \leq |s| < 2 \times 10^{-3}$), moderately deleterious ($2 \times 10^{-3} \leq |s| < 2 \times 10^{-2}$), and strongly deleterious ($2 \times 10^{-2} \leq |s| \leq 1$).

For each DFE and dominance model, we ran 25 simulation replicates and, at the conclusion of the simulation, outputted the mean genetic load, mean inbreeding load, and average number of derived deleterious alleles per individual in each population, taken from a sample of 100 individuals. Here, we define genetic load as the reduction in fitness due to segregating and fixed deleterious alleles, where fitness is multiplicative across sites [5,35,76]. We measure the inbreeding load as the "number of haploid lethal equivalents", which quantifies the summed selective effects of heterozygous recessive deleterious alleles in a population [4,35,46]. Due to the high computational load of these simulations, we ran simulations for 11 autosomes and projected results to a full genome of 22 autosomes. To do this, we exponentiated the outputted relative fitness values by 2 (subtracting this from 1 to obtain genetic load) and multiplied the outputted inbreeding load and derived allele counts by 2 (as these quantities are summed across chromosomes). Note that this procedure results in the simulation variance being overestimated, though the averages of these quantities are unbiased.

## Supporting information

**S1 Table. Demographic model parameter estimates from the synonymous site frequency spectrum.** Parameters are as follows: $N_{chrm}$ denotes the number of haploid chromosomes in the sample of the European ancestry individuals in the 1000 Genomes Project; $\theta_S$ is the mutation rate at synonymous sites; $N_1$, $N_2$ and $N_C$ are the population sizes relative to the ancestral

population, and $T_1$, $T_2$, and $T_C$ are the length of time in each epoch in $2N_{anc}$ generations.
(XLSX)

**S2 Table. Population genetic parameters used to scale the DFE in terms of *s*.** Parameters are as follows: $N_{chrm}$ denotes the number of haploid chromosomes; $\theta_S$ is the mutation rate at synonymous sites; $L_{NS}/L_S$ is the ratio of possible nonsynonymous to synonymous mutations; μ is the per base pair mutation rate per generation; $L_S$ is total count of synonymous sites, $L_{NS}$ is the total count of nonsynonymous sites; $N_{anc}$ denotes the ancestral population size.
(XLSX)

**S3 Table. Inference of the DFE assuming different dominance coefficients under a gamma distribution.** E[s] represents the expected selection coefficient, which is computed as $\alpha^*\beta/N_{anc}$. AIC refers to the Akaike information criterion based on model log-likelihood and number of parameters.
(XLSX)

**S4 Table. Fit of the discrete DFE assuming different dominance coefficients.** AIC was computed based on the number of free parameters of the discrete model (4 parameters). The *ΔAIC* corresponds to the difference between the corresponding gamma AIC to discrete models assuming the same dominance coefficient. Negative values indicate a better fit for the discrete DFE compared to the gamma DFE for that particular dominance coefficient (*h*).
(XLSX)

**S5 Table. Site frequency spectrum for synonymous variants in the 1000 Genomes Project dataset.** The first entry is the number of singletons, followed by the number of doubletons, etc. Note, this SFS has not yet been folded.
(TXT)

**S6 Table. Site frequency spectrum for nonsynonymous variants in the 1000 Genomes Project dataset.** The first entry is the number of singletons, followed by the number of doubletons, etc. Note, this SFS has not yet been folded.
(TXT)

**S7 Table. Log-likelihoods and attributes of 4096 DFE/*h* models fit to the 1000 Genomes Project SFS.** First column shows the log-likelihood of each model. Columns 2–6 show the MLEs of the proportions of mutations inferred in each bin of the DFE. Columns 7–11 show the value of *h* assumed for the 5 bins of the discrete DFE. Columns 12–16 show the expected value of *s* for each bin of the DFE, multiplied by the corresponding value of *h*. "average_inferred_hs" corresponds to the average *h*\**s* inferred for each model, where each bin of the DFE is weighted by the proportion of mutations inferred to be in that bin. This was plotted in S7 Fig. "mean_h" is the average of *h* across all 5 bins of the DFE. "mean_hs" is the average of Columns 12–16 for each DFE. This is the average *h*\**s* model space shown in S8 Fig "neut" corresponds to $0<|s|\leq10^{-5}$, "nneut" corresponds to $10^{-5}<|s|\leq10^{-4}$, "wkdel" corresponds to $10^{-4}<|s|\leq10^{-3}$, "moddel" corresponds to $10^{-3}<|s|\leq10^{-2}$, and "strdel" corresponds to $10^{-2}<|s|\leq0.5$.
(XLS)

**S1 Fig. Comparison of site frequency spectra for the gamma DFE model for different values of *h*.** Results shown for each *h* value relative to the empirical SFS from the 1000G dataset.
(PDF)

**S2 Fig. Comparison of site frequency spectra for the discrete DFE model for different values of *h*.** Results shown for each *h* value relative to the empirical SFS from the 1000G dataset.
(PDF)

**S3 Fig. Performance of the *h* and *s* inference approach on simulated data.** Dominance parameters (left column), selection parameters (middle column), and model fit (right column) for the true model are shown in orange. Note that the true model was 2.54 LL units away from the MLE and that the high LL models with monotonic decay (bottom row) have qualitatively similar dominance and selection parameters to the true model.
(PDF)

**S4 Fig. The relationship between *h* and *s* inferred from *Arabidopsis* does not fit the human nonsynonymous SFS.** Top: Comparison of dominance parameters for models considered, including a fully additive model, a model with a monotonic decay in *h* that is within 1.92 LL units of the MLE, and the model from Huber et al. [25] estimated for *Arabidopsis*. Middle: Model fit in comparison to the additive model. Note that the additive model and monotonic models have similar log-likelihoods though the Huber et al. [25] model has a much worse log-likelihood. Bottom: Discrete DFE parameters estimated for each dominance model when fit to the human nonsynonymous SFS. *Y*-axis shows the proportion of mutations in each bin of the discrete DFE.
(PDF)

**S5 Fig. Comparison of the SFS between models where *h* is fixed for a given *s* to models where *h* can vary.** Both models assume a Strongly Recessive model (**Table 2**), however, in the 'variable *h*' model, *h* is allowed to vary for a given bin of *s*. Note that the SFS for fixed and variable *h* models are both quite similar and closely match the empirical 1000 Genomes Project data.
(PDF)

**S6 Fig. Simulation results comparing predicted genetic load, inbreeding load, and derived allele count for African (AF) and European (EU) populations under a model where *h* can vary for a given *s*.** (A) Predicted genetic load in African and European populations. (B) Predicted inbreeding load in African and European populations. (C) Predicted derived deleterious allele count in African and European populations. Results are shown as boxplots summarizing output from 25 simulation replicates under each model. Note that the fixed *h* results on the left are from the Strongly Recessive model shown in **Fig 4**.
(PDF)

**S7 Fig. Inferred average *h\*s* for each of the 4096 discrete DFE/*h* models considered.** Inferred average *h\*s* is calculated as the sum over all the bins of the DFE of the expected value of *s* for the bin multiplied by the value of *h* for that bin multiplied by the proportion of mutations inferred to be in that bin of the DFE. Each point represents a particular model. Red points denote those models with a log-likelihood <4.74 units below the fully additive model. These models fit the data well. (A) All models. (B) Zooming in to the models with the highest log-likelihood.
(PDF)

**S8 Fig. Examination of the average *h\*s* model space.** The average *h\*s* space is calculated as the average over the 5 bins of the discrete DFE of the expected value of *s* for each bin multiplied by *h* for that bin. (A) Average *h\*s* values of the 4096 models evaluated. Note that the space of average *h\*s* is not exactly uniform. (B) However, models across the range of the average *h\*s* space have high log-likelihood, indicating a good fit to the data. Red points denote those models with a log-likelihood <4.74 units below the fully additive model. (C) Same as (B), but zooming in on the top of the *y*-axis.
(PDF)

## Acknowledgments

We are grateful to Maria Izabel Cavassim Alves for her instrumental role in the early stages of this project and to Bernard Kim for assistance with Fit$\partial$a$\partial$i.

## Author Contributions

**Conceptualization:** Christopher C. Kyriazis, Kirk E. Lohmueller.

**Formal analysis:** Christopher C. Kyriazis, Kirk E. Lohmueller.

**Funding acquisition:** Kirk E. Lohmueller.

**Investigation:** Christopher C. Kyriazis, Kirk E. Lohmueller.

**Methodology:** Christopher C. Kyriazis, Kirk E. Lohmueller.

**Project administration:** Kirk E. Lohmueller.

**Resources:** Kirk E. Lohmueller.

**Software:** Christopher C. Kyriazis, Kirk E. Lohmueller.

**Supervision:** Kirk E. Lohmueller.

**Visualization:** Christopher C. Kyriazis, Kirk E. Lohmueller.

**Writing – original draft:** Christopher C. Kyriazis, Kirk E. Lohmueller.

**Writing – review & editing:** Christopher C. Kyriazis, Kirk E. Lohmueller.

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
