## [Decision Letter · Decision Letter 0]

29 Apr 2024

Dear Dr Lohmueller,

Thank you very much for submitting your Research Article entitled 'Constraining models of dominance for nonsynonymous mutations in the human genome' to PLOS Genetics.

The manuscript was fully evaluated at the editorial level and by independent peer reviewers. The reviewers appreciated the attention to an important problem, but raised substantial concerns about the current manuscript. Based on the reviews, we will not be able to accept this version of the manuscript, but we would be willing to review a much-revised version. We cannot, of course, promise publication at that time.  To be considered again at PLOS Genetics, you will need to fully address the conceptual and statistical concerns that the reviewers raise. We expect this will require additional analyses. 

If you decide to revise the manuscript for further consideration at PLOS Genetics, please aim to resubmit within the next 60 days, unless it will take extra time to address the concerns of the reviewers, in which case we would appreciate an expected resubmission date by email to plosgenetics@plos.org.

We are sorry that we cannot be more positive about your manuscript at this stage. Please do not hesitate to contact us if you have any concerns or questions.

Yours sincerely,

Takashi Gojobori

Academic Editor

PLOS Genetics

Kelly Dyer

Section Editor

PLOS Genetics

Reviewer's Responses to Questions

**Comments to the Authors:**

Reviewer #1: In their manuscript, “Constraining models of dominance for nonsynonymous mutations in the human genome”, Kyriazis and Lohmueller used a combination of modelling and simulations with publicly available human genome data to assess the fit of various dominance and selection models for nonsynonymous mutations in human populations as well as examine how dominance affects the relative genetic load burden of deleterious variation in ancestral African versus derived European human populations. The authors found that a wide range of h values could fit the data, a negative relationship between h s, and that relative genetic load burden in the non-African population was dependent upon the dominance model. The manuscript is clearly written and the topic itself is a timely and interesting contribution to the field. I have only a few comments that I think need addressing, which I have outlined in the major and minor points below:

Major:

1. The authors used 432 individuals with European ancestry (described in lines 382–4), however, it is unclear where these individuals came from. Are they from the same or different populations? If it is the latter, did the authors check for potential population substructure (or is anything on this published), which might affect their results?

Minor:

1. Line 41 – At the beginning of the line, should “heterozygous” be “homozygous”?

2. Lines 134–153: There seem to be some typos in this sentence: “Given that we cannot reliably separate s and h for based on genetic variation data, we instead constrain the range of dominance models are consistent with the nonsynonymous SFS.” It is hard to understand what the authors mean.

Reviewer #2: Summary

It is generally accepted that dominance and selection coefficients are only weakly identifiable, such that a range of h values may be consistent with frequencies of deleterious alleles. It is not known how wide this range is and how it varies with s, and this is the knowledge hole the authors attempt to fill. This goal and the Poisson random field DFE inference methods used to achieve it are appropriate. However, the parameterization of the DFE makes interpretation difficult and may have serious statistical issues. In addition, the potential for dominance to vary for a given strength of selection is not considered or mentioned, making the authors conclusions less broad than would be desired.

Major comments

Analyses of non-synonymous variation in humans typically assume additivity of selection coefficients (h=0.5) between heterozygous and homozygous genotypes. Simultaneously, it is appreciated that the identifiability of moderate variation in h is limited from allele frequency data. The authors note that this implies there is a range of h and s combinations that should be compatible with the distribution of allele frequencies. The aim of this manuscript is to determine where the boundaries of that compatible h-s space are and to provide a sense for the consequences of such a plausible departure from additivity. To this end the authors fit a large set of models to the distribution of missense variant frequencies from nearly all protein coding genes in European ancestry individuals in 1000 Genomes. Models with similar likelihoods to the best fit are considered to be plausible models of dominance. Using this approach, they report that a global h as low as 0.15 is consistent with the data and that h can be as low as 0.05 for strongly selected alleles.

I agree with both the general need for such a study and the general approach taken by the authors. Having a sense of the relevant parameter space would be a great resource for researchers looking to relax additivity assumptions. However, I find two serious issues with the manuscript in its current state. The first is that parameterizing and presenting presenting models in terms of h and s (the selection coefficient against homozygotes) leads to conceptual and statistical problems. As the authors note, the shape of the SFS is largely determined by h*s, the heterozygote selection coefficient. The distribution on h*s is the foremost object being fit by DFE inference, and is what most people think of as the DFE (due to h=0.5 assumption). The authors treat the distribution on s as the DFE. This can be confusing, and it also makes any plot of DFE changes difficult to interpret since it isn’t clear what differences are due to dominance versus an attempt to keep the h*s distribution relatively constant.

A potentially larger statistical problem with the (h,s) parameterization arises in the context of discrete DFE models. These models assume a uniform distribution on s within approximately each factor-of-ten range from s=1e-5 upwards. The relative weights of these bins are fit, along with a corresponding h, during the inference procedure. Allowing h to vary for each bin means that the model space of h*s distributions changes as well. For instance, holding all other bins at h=0.5, decreasing h for the first bin will create a gap between the first and second bins in h*s space. In general, changing h values creates gaps and overlaps which affect what h*s distributions are possible. It is therefore not clear whether likelihood differences between models result from aspects of the SFS informative about dominance, or whether they are due to changes in the h*s model space. Since all the main results rely on discrete DFE models, it isn’t clear whether the conclusions of the manuscript are robust. Repeating the analysis by putting the bins on h*s instead of s would likely solve this problem, though other robustness checks are possible.

The second major issue is the lack of consideration for or mention of the possibility that dominance may differ among mutations with the same s. This seems to be exactly the scenario suggested by human Mendelian genetics. If we consider genes causing Mendelian disorders to be in the same bin of fitness effects, both h=0 and h=1 examples clearly exist. Identifying loci, genes, and gene sets with dominance effects on traits or fitness has been a major focus of previous work in this field (see Palmer et al. 2023 Science and Balick et al. 2022 AJHG for two recent examples). Huber et al. 2018 Nat Comm did suggest that deterministic h-s models capture the average relationship when h is random, but isn’t clear that the average h|s is what one should care about for the purposes of the manuscript. If so, this should be justified. Is it the proportion of strongly recessive mutations or the average h that determines inbreeding load, for instance. What information about h|s is identifiable or possible to bound? The authors should address whether their conclusions are robust to variance in h.

Minor issues

When defining h in the introduction, a definition of s as specifically the homozygous selection coefficient should also be given.

Balick et al. 2022 AJHG should also be in the introduction as a study that used similar SFS methods without the demography or selfing tricks, and reported evidence for strong recessive selection on some gene sets.

84: It would be nice to expand briefly on what previously used ad hoc dominance models have been used given that this is specifically what the authors propose to improve upon.

146: I think it is fine, but the use of 1000 Genomes European data should be justified in a world where largely and more ancestrally homogeneous samples are available.

165: When describing the discrete DFE in the main text, it should be stated that s is uniform with each bin.

Reporting model fits using log likelihoods in the main text is somewhat difficult to read given that each instance contain 6 digits. The authors should consider ways to simplify this such as distance from a best-fitting additive model.

172: If the fit of the additive discrete DFE is slightly worse than gamma, how do we know that any changes to h are not, in some sense, trying to make the discrete DFE closer to the better-fitting gamma? In general, it would be good to give a sense for what frequency range is sensitive to changes in h or some other way to inspect model fit aside from raw likelihoods. The supplemental figures give some sense but are difficult to interpret.

184: Justify why allowing h to vary among different classes is more important than allowing h to vary generally.

Figure 1A: y-axis is dominated by h=0 in a way that makes most of it hard to read.

Figure 1B/D: It seems most likely the the DFE is changing in a way that makes the h*s distribution relatively constant. It would be natural to show the implied h*s distribution here instead.

221: Can the authors make any statement about whether monotonic decay models provide a better fit generally than models without this property?

234: Can the model averaging approach be used to provide a range of plausible h values rather than point estimates? Ranges seem more natural to the idea of constraining model space.

253: What is the justification for using forward simulations when such quantities can be calculated in the diffusion approach?

Figure 4: Boxplots appear to show significant variation among simulation replicates and are largely overlapping for the AF and EU populations. If that is correct, the comparisons of point estimates around line 280 in the text is misleading. A statistic like the “probability B_AF>B_EU” over possible evolutionary histories might be more appropriate.

305: The authors should be clear about in what sense this h~0.35 is an average.

Reviewer #3: This work attempts to disentangle the selection (s) and dominance (h) coefficient of a mutation, addressing an important and challenging aspect in genetics. It does so by fitting various models of selection and dominance to patterns of human genetic variation. The constrained models that best fit the data are then used to investigate if the Out-of-Africa bottleneck lead to differences in genetic and inbreeding load between Africans and non-Africans.

In their approach, the authors use the synonymous SFS to infer the demography and the nonsynonymous SFS to infer the DFE. Given that background selection might affect synonymous sites, the authors should justify their choice, compared to for instance inferring the demography based on the “neutral” part of the genome as defined in other studies (e.g., Pouyet et al., eLife, 2018). Replicating the results based on demographic inferences from a more reasonably “neutral” SFS would substancially improve this work. Alternatively, a cautionary note pointing to possible biases in the results emerging from biased demographic inferences should be provided.

The demographic inferences presented in this work are based on a pooled sample of Europeans. It is unclear if population structure within this sample can lead to the inference of parameters that are not suitable for human populations and how that could impact the results and conclusions. Does the range of dominance and selection coefficients hold for single populations?

Could the rejection of highly recessive models (h<0.05) be due to the restriction in the upper range of s? Can the authors provide an intuition for why lower values are not fitting the data? The finding of a lower bound of h for strongly deleterious mutations should be further discussed in light of previous findings.

The results from Figure 3, show that the model fit is only slightly improved when h=0.05 relative to the fully additive model. This seems to point to a general lack of power to distinguish models (also seen in Figure 2 by the large range of models having a log-likelihood<1.92 units from that of the MLE). Predictive simulations to measure the power of this approach to disentangle h and s would be helpful to understand the robustness of the results.

Minor comments:

I have identified a number of sentences who either lack or have some extra words (e.g., line 134-135, 253-255, and 296-298). I recommend a careful inspection and correction of the text.

Please specify which parameter is shown in the y-axis of the bottom panel of Figure S3.

**Have all data underlying the figures and results presented in the manuscript been provided?**

Reviewer #1: Yes

Reviewer #2: Yes

Reviewer #3: **No: **The SFS for the 1000G project data is shown in Figures S1 and S2 and could be provided in spreadsheet form

PLOS authors have the option to publish the peer review history of their article (what does this mean?). If published, this will include your full peer review and any attached files.

Reviewer #1: No

Reviewer #2: No

Reviewer #3: No

---

## [Decision Letter · Decision Letter 1]

4 Sep 2024

Dear Dr Lohmueller,

We are pleased to inform you that your manuscript entitled "Constraining models of dominance for nonsynonymous mutations in the human genome" has been editorially accepted for publication in PLOS Genetics. Congratulations!

Yours sincerely,

Takashi Gojobori, PhD

Academic Editor

PLOS Genetics

Kelly Dyer

Section Editor

PLOS Genetics

Comments from the reviewers (if applicable):

Reviewer's Responses to Questions

**Comments to the Authors:**

Reviewer #1: In their manuscript, “Constraining models of dominance for nonsynonymous mutations in the human genome”, Kyriazis and Lohmueller used a combination of modelling and simulations with the publicly available 1000 Genomes Project human genome data to assess the fit of various dominance and selection models for nonsynonymous mutations in human populations as well as examine how dominance affects the relative genetic burden of deleterious variation in ancestral African versus derived European human populations. The authors found that a wide range of h values could fit the data and that relative genetic load burden in the non-African population was dependent upon the dominance model. This manuscript is a resubmission that addresses a topic that is both a timely and interesting contribution to the field. The authors have addressed all of my previous concerns. I think they have also done a nice job of addressing the other reviewers’ concerns.

Reviewer #2: I appreciate that the authors have seriously engaged with the numerous suggestions I made in my review. I am largely convinced that any inelegance in parameterizing the DFE under dominance has not strongly affected results. The application of the procedure to simulated data nicely demonstrates that it is not doing anything too weird. The added section on the impact of variable h|s has also meaningfully improved the manuscript, in particular, the added Figure S6.

As a side note, it is absolutely possible to calculate the genetic and inbreeding load under a diffusion model. For examples see Gravel (2016) Genetics and Takou et al. (2021) MBE. The reasons for this are that the integrals \\int x \\tau(x|S) and \\int x^2 \\tau(x|S) converge (tau being the PRF SFS) and that load measures are additive. The benefit here of doing simulations in SLiM seems to be to get closer to the reality of linked selection. This is great but computationally costly and making diffusion calculations from dadi output would have allowed a fuller exploration of DFE parameter space.

Reviewer #3: The authors carefully addressed all issues that I have raised, and followed the suggestion to perform simulations to test the robustness of their inference approach. Overall, the additional analyses and information included in the manuscript helped strengthen their conclusions and clarified some metodological choices. I have no further questions or requests. This work provides a good reference for further studies on dominance and will be helpful for chosing parameters in models of deleterious variation.

**Have all data underlying the figures and results presented in the manuscript been provided?**

Reviewer #1: Yes

Reviewer #2: Yes

Reviewer #3: Yes

PLOS authors have the option to publish the peer review history of their article (what does this mean?). If published, this will include your full peer review and any attached files.

Reviewer #1: No

Reviewer #2: No

Reviewer #3: No

**Data Deposition**

http://datadryad.org/submit?journalID=pgenetics&manu=PGENETICS-D-24-00229R1

**Press Queries**

---

## [Editor Report · Acceptance letter]

16 Sep 2024

PGENETICS-D-24-00229R1 

Constraining models of dominance for nonsynonymous mutations in the human genome 

Dear Dr Lohmueller, 

We are pleased to inform you that your manuscript entitled "Constraining models of dominance for nonsynonymous mutations in the human genome" has been formally accepted for publication in PLOS Genetics! Your manuscript is now with our production department and you will be notified of the publication date in due course.

With kind regards,

Anita Estes

PLOS Genetics

On behalf of:
